# Assessing Regional Economic Performance in the Southern Thailand Special Economic Zone Using a Vine-COPAR Model

**Arisara Romyen** [1,2,*]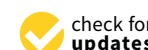**, Jianxu Liu** [3,4]**, Songsak Sriboonchitta** [2,4]**, Parinya Cherdchom** [1] **and Paratta Prommee** [1]

[1]  Faculty of Economics, Prince of Songkla University, Songkhla 90110, Thailand; parinya.c@psu.ac.th (P.C.); parata.p@psu.ac.th (P.P.)
[2]  Faculty of Economics, Chiang Mai University, Chiang Mai 50200, Thailand; songsakecon@gmail.com
[3]  Faculty of Economics, Shandong University of Finance and Economics, Jinan 250000, China; liujianxu1984@163.com
[4]  Puey Ungphakorn Center of Excellence in Econometrics, Faculty of Economics, Chiang Mai 50200, Thailand
*   Correspondence: arisara.r@psu.ac.th; Tel.: +66-89-729-0108

**Abstract:** Special economic zones (SEZ) can play an integral role in enhancing both regional and national economic growth. To explore the relationship between regional growth and the presence of an SEZ in Songkhla province, Thailand, the CD Vine–Copula AutoRegressive (CD-Vine COPAR) models were constructed using annual datasets of Songkhla's economic performance from 1995 to 2016. The findings indicate that the D Vine-COPAR model produced better fitting predictions for the manufacturing sector, while the C Vine-COPAR models better fit for the agriculture and service sectors. A five-year forecast (2017–2021) was also created. For Vine-COPAR-based Granger causality, the Gross Provincial Production, Foreign Direct Investment and Border Trade are evidently important contributors to regional economic development. Consequently, the government should adopt comprehensive strategies to ensure comparative advantages for operating in the region based on favorable local factors.

**Keywords:** Vine copulas; multivariate time series; forecasting techniques; evaluation of prediction model; Granger causality

## 1. Introduction

Under the new growth theory (Romer 1986; Lucas 1988), endogenous growth models emphasize technical progress (capital stock, human capital and innovation). From the early 1990s, foreign direct investment (FDI) has been a crucial source of enhancing economic growth in emerging economies. Thailand's regional development has been enhanced by the development of a Special Economic Zone (SEZ), which is viewed as an effective economic instrument to accelerate economic growth. Since 1961, the SEZ development policies have been implemented by developing major infrastructure projects, and regional development programs. The World Bank (2017) states that creating an SEZ can stimulate economic expansion both within and outside the zone. Inside the zone, it attracts foreign investment as well as facilitates skills and innovation transfer, while outside the zone it evokes synergies and knowledge spillovers to foster additional economic activity. Typically, the success of an SEZ depends on the SEZ's characteristics, the structure and the regional and country contexts. Since an SEZ operates as a geographic region within a country, localized employment churn is registered as job creation.

Moreover, economic dynamism can account for variations in regional productivity, and improves the standard of living of the nation (Slaper 2014).

In 2014, the Thai government launched pilot SEZ projects in five border provinces, namely Tak, Aranyaprathet, Mukdahan, Songkhla, and Trat, in which the government granted investors special privileges including tax and non-tax incentives. The Gross Provincial Production (GPP) accounts for the chain volume measures of production outputs at regional and provincial levels. The compilation of GPP as part of the system for National Accounts of Thailand is performed using a bottom-up approach, which effectively compiles and improves the indicators of provincial-level production. There are 16 economic activities including agriculture production (agriculture, hunting and forestry; and fishing) and non-agriculture production (mining and quarrying; manufacturing; construction; sale; hotels and restaurants; financial intermediation; real estate, renting and business activities; etc.). The change in real value added is employed in evaluating the changes of each production output. Looking more closely at economic performance through GPP, Songkhla produces many agricultural products (14% share of GPP), which provides copious raw materials, thus encourages agro-industries such as rubber, palm oil and seafood processing (21% share of GPP). Songkhla has a strategic location on the North–South Economic Corridor, which is directly connected to Malaysia and Singapore. It is a service-based economy (18% share of GPP) (Government Public Relations Department 2016). The values of cross-border trade account for more than 50% of Thailand's total border trade. Therefore, the major sectors contributing to economic growth can be categorized into three groups: (1) manufacturing; (2) agriculture; and (3) service sector (Cherdchom et al. 2016). As a result, forecasting and measuring of the GPP, FDI and border trade are crucial for driving the SEZ operation.

Generally, development is measured using aggregated models based on centralized data. However, a top-down framework can frequently be ineffective at a local level. In fact, regional performance can be better measured to explore economic convergence based on specific local characteristics (Ascani et al. 2012; Pan and Ngo 2016). Many empirical studies on economic growth dependence employ Granger's theorem (Engle and Granger 1987) or vector autoregressive (VAR) models (Hamilton 1994; Tsay 2002; Lütkepohl 2005). However, classical VAR models can capture only linear and symmetric dependences in time and between series. In a small degree of freedom, if the dependences of the system are modeled with longer lags, those larger estimated parameters might be misspecified. It is a weakness of the traditional VAR models (Pindyck and Rubinfeld 1997). Unbiasedness is a desired property of an estimator. Recently, research into the relationship between regional growth and SEZ remains limited (Lau 2010; Ho 2004). To address this issue, Vine–Copula autoregressive (Vine-COPAR) models are constructed to overcome such limitations as constants in dependency. The Vine-COPAR models have been verified as a flexible model with high-dimensional patterns (Brechmann and Schepsmeier 2013; Brechmann and Czado 2015). This approach allows arbitrary marginal distributions. To average interdependencies among multivariate time series, we exploited a fully integrated Granger causality test of Vine-COPAR models.

The objectives of this study were as follows. (1) We constructed so-called Vine-COPAR models. More precisely, we tested the Vine pair-copula decompositions including Canonical vines (C-vine) and Drawable vines (D-vine) to optimally quantify the dependence structures. (2) We evaluated the performance of the prediction methods. (3) We forecasted the SEZ's economic performance over the next five years (2017–2021). (4) We analyzed the patterns of causality among economic growth, FDI and border trade. The contributions of the paper are twofold. First, it is the first study to apply Vine-COPAR to analyze the dependence between economic growth and SEZ operation. Second, it is shown that adopting appropriate policies regarding GPP, FDI and border trade can provide new insights to highlight opportunities for competitive advantages within regional economy. This paper is structured as follows. Section 2 introduces the C and D Vine copulas, and the Vine-COPAR-based Granger causality. Section 3 presents data and empirical findings. Finally, Section 4 offers conclusions along with policy recommendations.

## 2. Methodology

### 2.1. Vine Copula Models

A copula function can be used to model multivariate distributions with given univariate margins. For a random vector $X = (X_1, \ldots, X_d) \sim H$ along with $H_i$, $i = 1, \ldots d$, Sklar's theorem (Sklar 1959) can be written as:

$$H(X_1, \ldots, X_d) = C(F_1((X_1), \ldots, F_d(X_d)), \tag{1}$$

where $H$ is an $n$-dimensional distribution with marginal $F_i$, $i = 1, 2 \ldots, d$. $C$ is a $d$-dimensional copula linked to form a joint distribution. Skalr's theorem with a bivariate copula is implemented as:

$$H(u_1, u_2) = C(F_1^{-1}(u_1), F_2^{-1}(u_2)), \tag{2}$$

where $u_1, u_2 \in [0, 1]$ and $F$ is the distribution of invertible margins $F_1$ and $F_2$.

C and D Vine Copulas

For multivariate analysis, C and D Vine copulas, which were initially proposed by Joe (1996), were utilized to estimate dependency. $X = x_1, x_2, x_3$ are the factors within the marginal distribution functions. For instance, C Vine copulas can be formed as Equation (3):

$$f(x_1, x_2, x_3) = f_1(x_1) f(x_2|x_1) f(x_3|x_1, x_2) \tag{3}$$

followed by Sklar's theorem (Equation (1)) such that

$$f(x_2|x_1) = \frac{f(x_1, x_2)}{f_1(x_1)} \tag{4}$$

$$f(x_2|x_1) = c_{1,2}(F_1(x_1), F_2(x_2)) f_2(x_2), \tag{5}$$

and

$$f(x_3|x_1, x_2) = \frac{f(x_2, x_3|x_1)}{f(x_2|x_1)} \tag{6}$$

$$f(x_3|x_1, x_2) = c_{2,3|1}(F(x_2|x_1), F(x_3|x_1)).c_{1,3}(F_1(x_1), F_3(x_3)) f_3(x_3). \tag{7}$$

### 2.2. VAR Model

Since the SEZ was considered as the economic transformation, the FDI and border trade were viewed as a proxy for the SEZ performance. Gross provincial production, foreign direct investment and border trade are denoted as GPP, FDI and TRADE, respectively. All relevant variables were transformed into logarithms to stabilize the variance of a series. The Kwiatkowski–Phillips–Schmidt–Shin (KPSS) test was used to test the stationary time series. We propose a VAR framework that consists of the three endogenous variables: $Y = (GPP, FDI, TRADE)$. The VAR ($p$)-process can be defined in the basic form as:

$$lnY_t = \beta_0 + \beta_i lnY_{t-i} + \ldots + \beta_p lnY_{t-p} + \varepsilon_t, \ t = p+1, \ldots, T, \tag{8}$$

where $\beta_0$ is a $k$-dimensional constant vector, $\beta_i$ are $k \times k$ real-valued matrices for $i = 1, \ldots, p$ and $\varepsilon_t$ is a $k$-dimensional white noise vector process with a time-variant definite covariance matrix $E(\varepsilon_t \varepsilon_t') = E_\varepsilon$ for $t = p+1, \ldots, T$.

We applied a VAR to forecast economic performance. The assumption of random repressors is uncorrelated with errors. For forecasting of the final period ($T$), vector $Y_t$ and $Y_{t-1}$ are essential. Pecican (2010) consigned a forecast for one-period as follow:

$$Y_{(T+1)/T} = E(y_{T+1}/y_T, y_{T-1}, \ldots) = \beta_1 y_T + \beta_p y_{T-p+1} \tag{9}$$

### 2.3. Granger Causality

Since the Gaussian assumption is constant variance, this method can merely capture linear and symmetric dependence in time and between series. Therefore, we employed the Vine-COPAR model based Granger causality, which determines interdependencies among multiple time series, to deal with high-order causality. The Vine-COPAR($p$)-based Granger causality was then built in the following form:

$$lnY_{kt} = \beta_{k0} + \sum_{i=1}^{s} \beta_{ki} lnY_{1t-i} + \sum_{j=1}^{q} \beta_{kj} lnY_{2t-j} + \sum_{l=1}^{n} \beta_{kl} lnY_{3t-l} + \varepsilon_{kt}. \tag{10}$$

where $\beta_{k0}$ (for $k = 1, 2, 3$) is the interception term. $\beta_{ki}$, $\beta_{kj}$ and $\beta_{kl}$ are the estimated coefficient of the lagged variables. $s$, $q$ and $n$ are the optimal serial lag lengths and $\varepsilon_{kt}$ refers to random disturbance terms.

The log-likelihood ratio (LR) tests of unrestricted ($L_u$) and restrictions ($L_r$) models were imposed with the null hypothesis of no Granger causality $H_0 : \beta_{ki,1} = \beta_{kj,2} \ldots = \beta_{kl,3} = 0$. Then, we employed the maximum likelihood method to maximize the Vine-COPAR models to obtain the estimated parameters, which was better than a two-step estimation.

## 3. Data and Empirical Results

In Section 3.1, the data are described and the empirical findings are reported in Section 3.2, including the accuracy of the prediction models, the forecast under the Vine-COPAR, and the Granger causality based on the Vine-COPAR models.

### 3.1. Data

The annual datasets of GPP and TRADE time-series data were obtained from the Bank of Thailand database, while the FDI data were acquired from the Thai Board of Investment over the period 1995–2016. Before that time, those statistical datasets were incomplete. The GPP was counted at chain volume measures (the reference year at 2002), the TRADE was measured at the FOB prices and the FDI was counted at current prices. Figure 1 shows Songkhla economic performance. The GPP has been increasing for the manufacturing and trade sectors, while the agriculture sector has been slowing down due to the agricultural price fluctuation especially in rubber, which is the most important cash crop in the south (Figure 1a–c). In 2005, Thailand Board of Investment started to approve inward Thai direct investment and FDI in southern Thailand. Afterwards, the FDI volumes seemed surge and stop in neighboring countries such as Vietnam and Indonesia, seeking cheaper material resources or labor offshore.

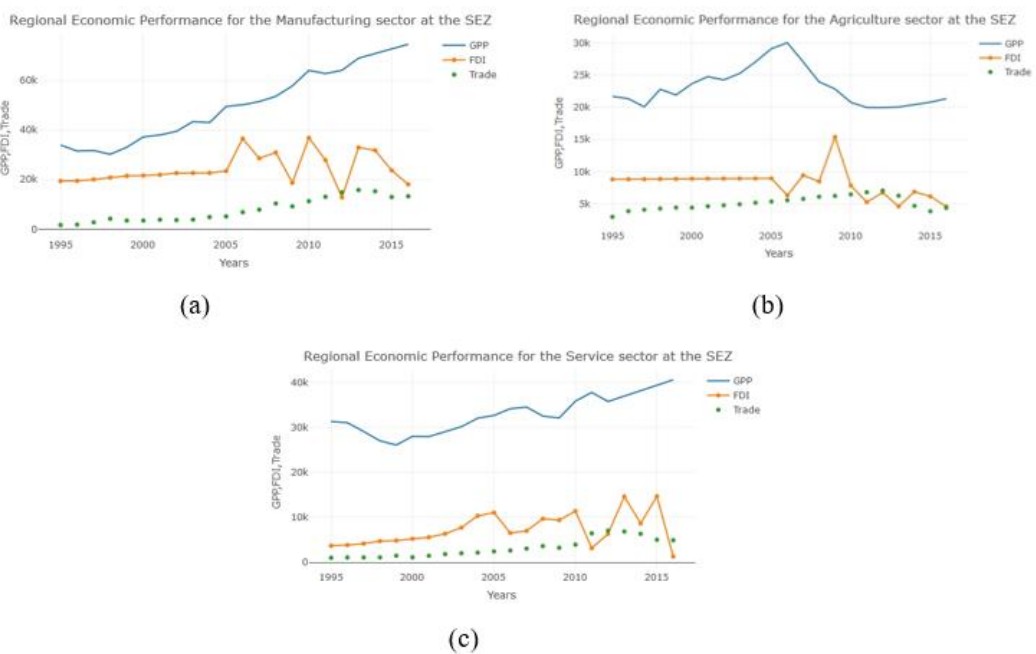

**Figure 1.** The Regional Economic performances in the Southern Thailand SEZ.

*3.2. Empirical Results*

3.2.1. CD-Vine COPAR Models

The dependence of economic data was explored within: (1) manufacturing; (2) agriculture; and (3) the service sector. Macro-economic time-series data are usually non-stationary (Nelson and Plosser 1982). Hence, the KPSS test was utilized within the null hypothesis of a stationary against a unit root alternative. All variables were stationary at an alpha level of 0.01 in all cases (for example, in the manufacturing sector for the GPP, it gives KPSS level of 0.166, truncation lag parameter of 2 and p-value of 0.121; for the FDI, it reports KPSS level of 0.168, truncation lag parameter of 2 and *p*-value of 0.152; and for the TRADE, it informs KPSS level of 0.199, truncation lag parameter of 2, *p*-value of 0.136). Consequently, the VAR specification in level was sufficient.

Table 1 summarizes that the dependencies between two variables belong to several copula families (Gaussian, Clayton, Frank, Joe, Rotated Joe, rotated Clayton, and Rotated Gumbel). The pair-copula constructions decompose a multivariate probability density towards bivariate copulas for a good selection choice to build more powerful tree sequences. We then informed the bivariate copula families serving as building blocks for vine copulas. For instance, the optimal pair-copula constructions based on the C-Vine COPAR between the GPP (1) and FDI (2), the GPP (1) and TRADE (3), and the FDI–TRADE (23) and GPP (1) in the manufacturing sector were Joe, rotated Joe and Frank families, respectively. The Kendall's tau reported both positive and negative dependencies. For tail dependencies in manufacturing sector (D-Vine COPAR), the GPP and FDI had a great dependence in the upper tail, indicating that GPP is likely to raise together with FDI; GPP and TRADE were similar. For the agriculture sector (C-Vine COPAR), the GPP and TRADE had dependence in the lower tail. Furthermore, we can see that the AIC to penalize copula families with more parameter for D Vine-COPAR model (59.125) produced a better performance in the case of the manufacturing sector, while the C Vine COPAR models (100.422 and 111.162) provided properly good fit models for the agriculture and service sectors. After choosing the independence copula, the optimal lag of the Vine COPAR(p*) models were D-Vine COPAR (3), C-Vine COPAR (3) and C-Vine COPAR (2) for the manufacturing, agriculture and service sectors, respectively.

**Table 1.** CD-Vine COPAR models, Kendall's correlation, and upper-lower tail dependence.

| SEZ Economic Performance | Manufacturing | | Agriculture | | Service | |
|---|---|---|---|---|---|---|
| | **C−Vine COPAR** | **D−Vine COPAR** | **C−Vine COPAR** | **D−Vine COPAR** | **C−Vine COPAR** | **D−Vine COPAR** |
| **pair−copula** [1] | 5.470 (12 = Joe) | 4.682 (12 = Joe) | −0.010 (12 = rotated Clayton) | −0.042 (12 = rotated Clayton) | 0.510 (12 = Gaussian) | −0.411 (12 = Gaussian) |
| | 1.221 (13 = rotated Joe) | −2.303 (23 = Frank) | 1.581 (13 = Rotated Joe) | −1.046 (23 = rotated Gumbel) | 0.486 (13 = Frank) | −1.024 (23 = Frank) |
| | −4.583 (231 = Frank) | 1.276 (132 = Clayton) | −16.812 (231 = Frank) | −0.164 (132 = Gaussian) | −0.954 (231 = Frank) | −0.388 (132 = Gaussian) |
| **Kendall's tau** | 0.700 | 0.660 | −2.516 | −0.020 | 0.340 | 0.221 |
| | 0.380 | −0.243 | 0.245 | −0.044 | 0.054 | −0.046 |
| | −0.430 | 0.389 | −0.823 | −0.105 | −0.410 | −0.267 |
| **tail dependence** [2] | [0,0.865] | [0,0.840] | [0,0] | [0,0] | [0,0] | [0,0] |
| | [0,0.567] | [0,0] | [0.450,0] | [0,0] | [0,0] | [0,0] |
| | [0,0] | [0.581,0] | [0,0] | [0,0] | [0,0] | [0,0] |
| **AIC** | 76.254 | 59.125 | 100.422 | 135.718 | 111.162 | 131.362 |
| **Vine−COPAR(*p*)** [3] | D−Vine COPAR *(3)* | | C−Vine COPAR *(3)* | | C−Vine COPAR *(2)* | |

Note: [1] pair-copula: GPP (1), FDI (2), TRADE (3); [2] tail dependencies: [lower, upper]; [3] selection for appropriate lags for manufacturing: AIC ($-5.25 \times 10^1$), HQ ($-5.35 \times 10^1$), SC ($-5.21 \times 10^1$), and FPE ($7.45 \times 10^{-23}$); for agriculture: AIC ($-5.24 \times 10^1$), HQ ($5.31 \times 10^1$), SC ($-5.26 \times 10^1$), and FPE ($7.24 \times 10^{-23}$); for service: AIC ($-5.15 \times 10^1$), HQ ($5.41 \times 10^1$), SC ($-5.27 \times 10^1$), and FPE ($7.18 \times 10^{-23}$).

### 3.2.2. Evaluation of the Performance of the Prediction Models

Since traditional VAR models can only handle linear and symmetric dependence structures, we developed novel CD-Vine COPAR models, which allowed for asymmetric modeling of serial and between-series dependencies to advocate superior predictive ability of the CD-Vine COPAR models over the classical VAR models for benchmarking. The prediction performance of the D-Vine COPAR model was compared to that of a classical VAR based model employing the out-sample (2012–2016). For evaluation of performance, the Root Mean Square Error (RMSE) and the Mean Absolute Error (MAE) were measured, as shown in Table 2. The CD Vine-COPAR had better prediction accuracy with smaller RMSE and MPE.

**Table 2.** Comparison of the predictive power of the CD-Vine COPAR, classical VAR and BOOTSTRAP with Vine COPAR.

| | Manufacturing | | Agriculture | | Service | |
|---|---|---|---|---|---|---|
| **Accuracy** | **D-Vine COPAR** | **Classical VAR** | **C-Vine COPAR** | **Classical VAR** | **C-Vine COPAR** | **Classical VAR** |
| **RMSE** | 0.679 | 0.685 | 0.998 | 1.024 | 1.140 | 1.145 |
| **MAE** | 0.480 | 0.482 | 1.106 | 1.219 | 0.808 | 0.829 |
| | **BOOTSTRAP with D-Vine COPAR** [1] | | **BOOTSTRAP with C-Vine COPAR** [1] | | **BOOTSTRAP with C-Vine COPAR** [1] | |
| **RMSE** | 0.674 | | 0.992 | | 1.134 | |

Note: [1] A positive integer giving the number of bootstrap replicates requires R = 25 and the type of simulation is the fixed block length used in generating the replicate time series at l = 20.

Furthermore, to ensure the statistical significance, we also conducted the block bootstraping procedure, which involved iteratively resampling the dataset with replacement to examine the statistical significance though the specific time periods. The number of bootstraping replications (R) was 25 and the replicated time series were simulated using the type of block resampling with fixed block lengths (l = 20) over the samples of 22 elements, obtaining the total number of 550 experiments. The bootstrap-based CDVine COPAR models were slightly better than the CD-Vine COPAR models and the classical VAR models with respect to the RMSE.

### 3.2.3. The Prediction of the SEZ's Economic Performance for the Next Five Years

It is questionable whether economic models represent accurate forecasting techniques. To consider this issue, the statistical test of a model's forecast performance was technically conducted by dividing a given dataset into the in-sample period used for initial parameter estimation, and the out-of-sample period used for evaluating forecasting performance. Empirical evidence relying on out-of-sample forecast performance is more trustworthy and better reflects the information in real time than evidence based on in-sample performance due to the sensitivity of outliers and the process of discovering patterns in the datasets (Ehling and Körner 2015). In this study, the in-sample data (1995–2011, roughly 70% of the data) were used to construct the estimation, while the out-of-sample data (2012–2016, the remaining 30%) were employed to produce the prediction. Afterwards, we forecasted the next five-year interval (2017–2021). The classical VAR versus Vine-COPAR models were compared using RMSE and MPE to evaluate their predictive power. Ultimately, the forecasts of the SEZ's economic performance for the next five years for the manufacturing, agriculture and service sectors were carried out. Figure 2 displays the GPP, FDI and TRADE performances throughout the 2017–2021 forecast period using D Vine-COPAR for the manufacturing sector, and C Vine-COPAR for the agriculture and service sectors. Interestingly, the FDI seemed to increase in value, especially in the manufacturing and service sectors, while TRADE stayed constant for the three sectors. The GPP and FDI flows exhibited high levels of volatility.

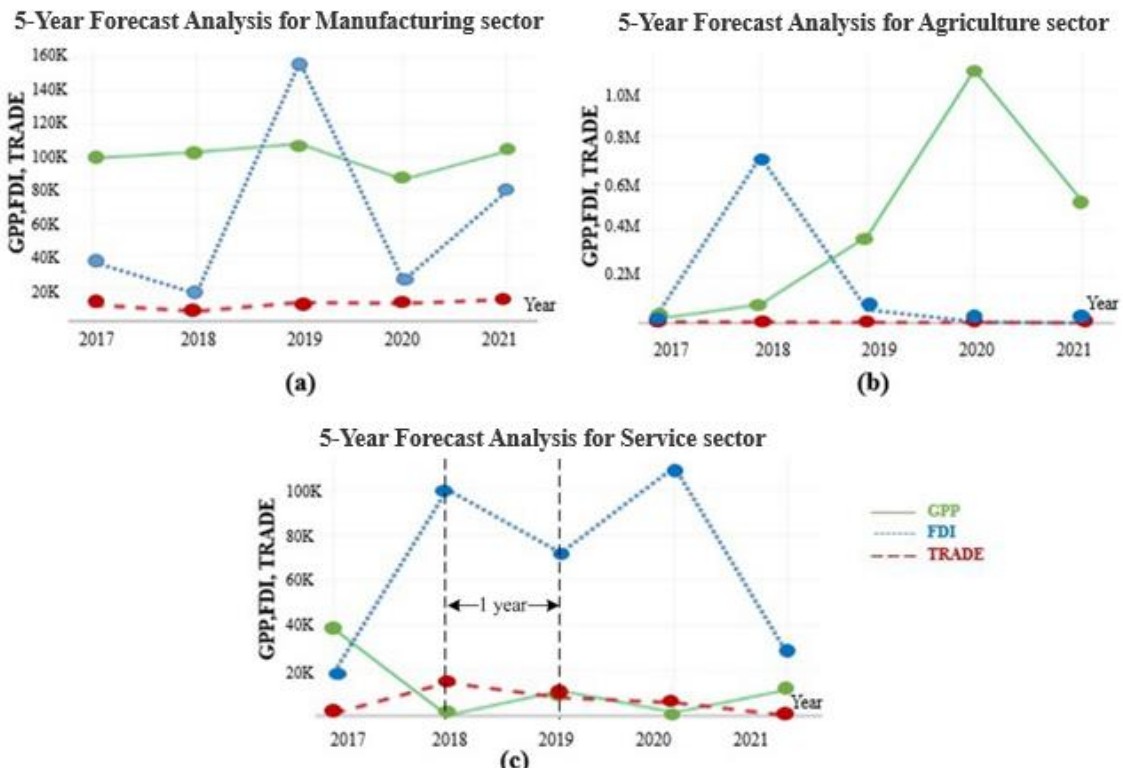

**Figure 2.** The forecasts of the SEZ's economic performance for the next five years.

### 3.2.4. Vine COPAR-Based Granger Causality

The results of the LR tests is shown in Table 3 under null hypothesis of non-causality. For the manufacturing sector, Granger-bidirectional causality existed between GPP and FDI, and TRADE was Granger-causal of FDI. Consistently with the agriculture sector, Granger-bidirectional causality occurred between FDI and TRADE, and GPP was Granger-causal of TRADE. In addition, for the service sector, Granger-bidirectional causality took places between GPP and FDI as well as GPP and TRADE, while FDI was Granger-causal of TRADE.

**Table 3.** Vine COPAR-based Granger causality test results.

| Null Hypothesis | Statistics | Manufacturing | Agriculture | Service |
|---|---|---|---|---|
| FDI does not Granger-cause GPP | LR test | 13.406 | 0.367 | 22.287 |
| | *p*-value | 0.004 *** | 0.947 | $5.683 \times 10^{-5}$ *** |
| Trade does not Granger-cause GPP | LR test | 0.361 | 4.368 | 24.438 |
| | *p*-value | 0.948 | 0.224 | $2.023 \times 10^{-5}$ *** |
| GPP does not Granger-cause FDI | LR test | 17.956 | 3.714 | 23.443 |
| | *p*-value | 0.000 *** | 0.294 | $3.265 \times 10^{-5}$ *** |
| TRADE does not Granger-cause FDI | LR test | 10.301 | 13.814 | 19.852 |
| | *p*-value | 0.016 ** | 0.003 *** | 0.000 |
| GPP does not Granger-cause TRADE | LR test | 2.924 | 34.975 | 23.205 |
| | *p*-value | 0.404 | $1.233 \times 10^{-7}$ *** | $3.660 \times 10^{-5}$ *** |
| FDI does not Granger-cause TRADE | LR test | 3.550 | 33.741 | 23.930 |
| | *p*-value | 0.314 | $2.247 \times 10^{-7}$ *** | $2.583 \times 10^{-5}$ *** |

Note: Significance codes are defined as *** = 0.01, ** = 0.05 and * = 0.1.

## 4. Conclusions and Policy Implications

This study examined the relation between economic growth and the SEZ in Songkhla province, including evaluating the accuracy of the prediction approach, forecasting the economic performance of the SEZ, and examining the causal influences among GPP, FDI and TRADE using Vine-COPAR based Granger causality. The main finding can be drawn as follows:

(1)　The appropriate specification for a forecasting method using a Vine-COPAR model provides better results than a single time series, since evaluating more dependence structures leads to more accurate predictions. Moreover, the Vine-COPAR based Granger causality can accommodate high-order moment causality and this approach thus provides effective long-run performance.

(2)　For five-year forecast (2017–2021), the FDI and TRADE appeared to be the important contributions towards the SEZ. However, GPP and FDI displayed sharp fluctuations, and TRADE behaved constantly. Therefore, the government should encourage their competitiveness and maintain continuity of foreign investment and trade policies.

(3)　Granger causality and bidirectional causality existed among GPP, FDI and TRADE in all sectors.

This study has some policy implications. The government should promote FDI-friendly policies and trade promotion in the SEZ since these policies can play a crucial role in boosting regional economic growth. The core of the development should focus on the favorable privileges towards FDI; in addition, the free trade regime should be supported by eliminating several non-tariff barriers, particularly bearing in mind that the FDI based-enterprises within SEZ are ordinarily accorded more liberal operating conditions.

Consequently, the findings from the provincial level suggest that Songkhla province should be promoted as one of the nine industrial centers in Thailand since it has great potential for a considerable acceleration of its economic growth based on the SEZ policy, which could contribute towards the enhancement of the regional economy.

**Author Contributions:** A.R. wrote the paper, conducted the research findings, and revised the manuscript. J.L. constructed the model analysis and source coding for the R software. P.C. and P.P. performed field interviews, collected data and analyzed the results. S.S. designed the overall research and revised the policy recommendations. All authors revised and approved the final manuscript.

**Funding:** This work was supported by the Office of the Higher Education Commission (OHEC), Thailand; the Research and Development Office, Prince of Songkla University; and the Faculty of Economics, Prince of Songkla University, Thailand under research grant ECO580922S.

**Conflicts of Interest:** The authors declare no conflict of interest.

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
