# Peer review of "Assessing Regional Economic Performance in the Southern Thailand Special Economic Zone Using a Vine-COPAR Model"

_economies, doi:10.3390/economies7020030_

Round 1
Reviewer 1 Report
Comments on Assessing Regional Economic Performance in the Southern Thailand Special Economic Zone using a Vine-COPAR Model
The paper explores the relationship between regional growth and the presence of the SEZ in Songkhla province, Thailand using CD Vine-COPAR models based on the annual data of Songkhla’s economic performance from 1995 to 2016. The findings indicate that the D Vine-COPAR model produced better prediction for the manufacturing sector, while the C Vine-COPAR models provided better goodness of fit for the agriculture and service sectors. The causal link between GPP, FDI and Trade are examined through Vine-COPAR model.
Specific Comments:
1. The authors use PP test to examine the unit root property of GPP, FDI and Trade variables. What is the outcome of the results? Are these variables stationary? Or non-stationary? If they are stationary, then VAR specification in level is appropriate. If they are non-stationary and they cointegrated then it can be estimated in VECM specification. On the other hand, if they are non-stationary and not-cointegrated then the VAR in difference form is sufficient. It is not clear to the readers if they are stationary or not. Why do you choose PP test as opposed to other alternatives? Why don’t you consider the stationary null (KPSS) as well to make sure that the results are not affected by the lack of statistical power.
2. In equation (1.8), both intercept and the coefficient of appears as . It should be fixed.
3. The intercept is assumed as a column vector of order Mx1 while the coefficient matrix is of order KxK. The matrix operation is not confirmable when M ≠ K.
4. The section 2.3 doesn’t make any sense. What do you mean by higher-order causality?
5. In lines 106 and 107, the operator and the parameter are missing. It appears as “denotes the first difference”. Similarly, “refers to random disturbance term”
6. The hypothesis in line 110 has no link to the working model represented by equation (1.10).
7. The results reported in Table 1 is not self-expletory and the discussion of results in section 3.2. is not clear with reference to table 1.
8. Seventeen years of annual data is used for estimation and then five years of data is used for forecast evaluation. It makes no sense to use small sample for estimation and then use the model for prediction.
9. The results illustrated in table 2 show that COPAR model is marginally better than classical VAR model. Are these differences statistically significant? It can be examined through block-bootstrapping. But again, it will not be beneficial given the small sample size. Why classical VAR model is used as benchmark model for comparison? Why don’t you compare the other alternative models?
Author Response
Please see more details for the amendment within a point-by-point response to the reviewer 1’s comments as the attached file.
Ps. Response to Reviewer 1 Comments is below added to the end of my manuscript.

Reviewer 2 Report
Moreover, there should be explained the method of calculation of gross provincial production (GPP).
Certainly the introduction requires a deeper theoretical background regarding determinants of regional economic performance.
Analyzing economic performance of Songkhla province, it is worth to refer to other five provinces mentioned in line 28 or to the average situation in Thailand.In Conclusions, the policy implications should refer only to the analyzed province and not extrapolated to the general situation in Thailand.Moreover, the English language needs quite extensive revision by a mother tongue proofreader and the paper needs some stylistic corrections (e.g. line 15 – double dots, line 28 comma instead of dot before in, line 53 dot before bracket is not needed, line 116 no dot at the end of the sentence).
Author Response
Please see more details for the amendment within a point-by-point response to the reviewer 2’s comments as the attached file.
Ps. Response to Reviewer 2 Comments is below added to the end of my manuscript.

Reviewer 3 Report
In the paper “Assessing regional economic performance in the southern Thailand Special Economic Zone using a Vine-COPAR model” the authors apply Vine-COPAR model and Granger causality to forecast Gross Province Product (GPP), Foreign Direct Investment (FDI) and Border Trade in Special Economic Zone of Thailand, distinctly for manufacturing, agriculture and service sectors.
Although the topic is of interest, many drawbacks highly limit the proposed analysis. In general:
1. Extensive editing of English language and style is required
2. Many equations and formulas are inaccurate
3. Data should be better explained (at current prices, constant,...)
4. In the graphs of the 5 years forecasts, intervals should also be shown
5. The use of Vine-COPAR models should be better supported and some more statistics presented (e.g. the order of COPAR)
In particular, with respect to 2, note:
a) line 72, there is confusion between n and d
b) in equation (1.8) and line 94, φt is wrong
c) equation (1.9) is wrong
d) check equation (1.10), the limits of the sum operator. Give in case some reference
e) in equation (1.9) the model is expressed in terms of Y, in equation (1.10) in terms of ΔlnY. Is the series stationary?
f) Line 110, Granger causality test is given in terms of φ or β?
Author Response
Please see more details for the amendment within a point-by-point response to the reviewer 3’s comments as the attached file.
Ps. Response to Reviewer 3 Comments is added to the end of my manuscript.

Round 2
Reviewer 1 Report
The revised version of this paper reads better than the previous version. The authors have made significant improvement.
I am still convinced about the last point.
The results illustrated in table 2 show that COPAR model is marginally better than classical VAR model. Are these differences statistically significant? It can be examined through block-bootstrapping. I suggest to conduct block bootstrapping to examine the statistical significance though you have small sample.
Author Response
Please see a point-by-point response to the reviewer 1’s comments is detailed as the attached file

Reviewer 2 Report
I am satisfied with the implemented improvements.
Author Response
It's no improvement since the Reviewer 2 has satisfied the previous the implemented improvements.